# STDM: Spatio-Temporal Diffusion Models for Time Series Analysis

## Abstract

Denoising diffusion models have emerged as a formidable method, consistently surpassing previous state-of-the-art benchmarks. However, a notable challenge in time series-related tasks like anomaly detection and forecasting is the conditioning for models to reconstruct inputs accurately or generate samples based on past time steps rather than producing entirely new samples. To address this, we introduce a novel technique that enhances the sampling capabilities of denoising diffusion models for time series analysis, namely Spatio-Temporal Diffusion Models (STDM). While recent methods fall short of mapping contextual neighborhood dependencies directly into the sampling of a noisy sample, we focus on guiding the forward process of the diffusion model. The degeneration of a sample is based on the idea that values of neighboring time steps are highly correlated. We benefit from this assumption by presenting a diffusion step-dependent convolutional kernel to capture spatial relations and a combined, correlated noise to degenerate the input. Our method can be integrated seamlessly into various existing time series diffusion models. We compare the results of anomaly detection and forecasting when using the traditional and our novel forward process. In our experiments on synthetic and real-world datasets, we show that an adaption of the forward process can be beneficial, as our approach outperforms diffusion models with the ordinary forward process in task-specific metrics, underscoring the efficacy of our strategy in enhancing time series analysis through advanced diffusion techniques.

## 1 Introduction

Time series analysis is a cornerstone of modern applications across a multitude of domains, including healthcare (Morid et al., 2023), climate modeling (Mudelsee, 2019), industrial manufacturing (Ali Nemer et al., 2022), and cyber security (Al-Ghuwairi et al., 2023). Deep learning-based models have demonstrated remarkable capabilities in discerning patterns and dependencies within multivariate time series data. These models excel in reconstructing signals to detect anomalies and predicting future timestamps. Crucially, these tasks often necessitate unsupervised training, as existing datasets frequently lack labeled data, or the output itself is a time series.

Time series analysis inherent different challenges: An accurate reconstruction of input data is paramount for *anomaly detection* (Chandola et al., 2009), as anomalous data cannot be generated during inference due to the training dataset predominantly consisting of nominal state data. Time series *forecasting* necessitates the model to learn the historical patterns of the time series to predict future time steps (Lim & Zohren, 2021). Some approaches for forecasting include a reconstruction of the past time steps (Kollovieh et al., 2023).

In recent years, Denoising Diffusion Probabilistic Models (DDPMs) (Ho et al., 2020) have garnered significant attention in generative tasks owing to their exceptional ability to produce high-quality samples. These models operate by progressively distorting an input with Gaussian noise, training a model to reverse this process by estimating the corruption at various levels.

A critical aspect of time series anomaly detection and forecasting is to ensure the model utilizes the time series as an input during inference rather than generating a new, realistic time series from pure Gaussian noise. Current diffusion models fall short of paying attention to the temporal patterns within the time series during the forward process and rely solely on incorporating conditions during training and sample generation.

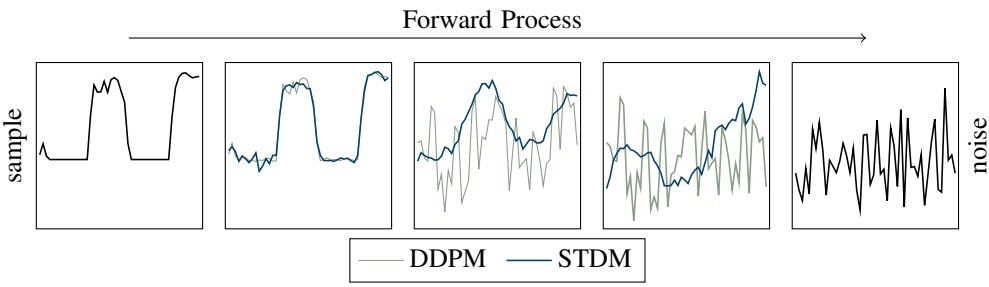

Figure 1: Difference between STDM and DDPM forward process at different diffusion steps. Both are using a linear scheduler and 100 diffusion steps. The original sample (left) is part of the `Solar` dataset.

In this paper, we propose Spatio-Temporal Diffusion Models (STDM), a novel technique that guides the diffusion model in reconstructing multivariate time series data by refining the forward process. Based on the idea of correlation neighboring time steps (Wu et al., 2021), our approach employs a convolution-based sampler that benefits from spatial relations within the time series while introducing a correlated noise to corrupt it to an unrecognizable state. Figure 1 shows the difference between STDM and the traditional DDPM forward process using a single channel of a time series. The mapped dependencies enable us to address task-specific challenges more effectively and to train more robust models. Our approach seamlessly replaces the standard diffusion sampling process and can be combined with various conditioning techniques. Adapting existing diffusion models for anomaly detection and forecasting offers distinct advantages over the traditional forward process when working with standard datasets.

We summarize our most important contributions as outlined below:

1. We introduce a novel diffusion forward process for time series data that takes spatial correlations into account while remaining a Markov chain.

2. Due to the similar structure as ordinary forward steps, our approach can be easily integrated into existing diffusion models.

3. We show that our approach can improve the results of time series diffusion models in anomaly detection and forecasting tasks.

The structure of this paper is as follows: Section 2 provides information on time series tasks, denoising diffusion probabilistic models (DDPMs), and approaches to conditioning and guidance. Our guided degradation process is detailed in Section 3. Section 4 demonstrates the effectiveness of our approach by modifying existing diffusion models. Finally, Section 5 summarizes the strengths and limitations of our approach.

## 2 PRELIMINARIES

### 2.1 PROBLEM STATEMENT

Let $x^0 \in \mathbb{R}^{d \times T}$ be a multivariate time series with a sequence length of $T$ and $d$ features at every time step. The index $0$ indicates that the data is in its uncorrupted, original form. The objective for a denoising network $\epsilon_\theta$ is task-specific for times series analysis.

**Anomaly detection.** In an anomaly detection task, non-normal time steps in $x^0$ should be detected. The network is trained to reconstruct the complete input, while the majority of the training data represent the nominal state. Therefore, anomalies cannot be reconstructed accurately. Depending on the evaluation strategy, a corresponding time step is considered abnormal if the reconstruction error surpasses a pre-defined or calculated threshold.

**Forecasting.** For time series forecasting, the values of the future time steps should be predicted. Given $x^0$, the model aims to continue the time series realistically for a definite amount of time steps. Depending on the approach, the model's output could also include a reconstruction of the

observation. Forecasting techniques can also be used for anomaly detection (Hundman et al., 2018) by comparing predicted and actual targets.

## 2.2 DIFFUSION MODELS

DDPMs are well-known diffusion models introduced by Ho et al. (2020). In the forward diffusion process, a sample $x^0 \sim q(x^0)$ is corrupted by gradually adding noise $\epsilon \sim \mathcal{N}(\mathbf{0}, \mathbf{I})$ to become a Gaussian noise vector $x^K$ at the final diffusion step $K$. This process can be described as the Markov chain

$$q\left(x^{1:K} \mid x^0\right) = \prod_{k=1}^{K} q\left(x^k | x^{k-1}\right) \tag{1}$$

with

$$q\left(x^k \mid x^{k-1}\right) = \mathcal{N}\left(x^k; \sqrt{1 - \beta_k}\ x^{k-1}, \beta_k \mathbf{I}\right) \tag{2}$$

with $\beta_k \in [0, 1]$ being the noise variance at diffusion step $k \in [1, K]$. It is possible to sample at any arbitrary step $k$ in a closed loop form with $\bar{\alpha}_k := \prod_{i=1}^{k} \alpha_i$, $\alpha_k := 1 - \beta_k$ and $\epsilon \sim \mathcal{N}(\mathbf{0}, \mathbf{I})$ as

$$x^k = \sqrt{\bar{\alpha}_k}\ x^0 + \sqrt{1 - \bar{\alpha}_k}\ \epsilon\ . \tag{3}$$

The backward denoising process starts with $x^k \sim q\left(x^k\right) = \mathcal{N}(\mathbf{0}, \mathbf{I})$ and can be described as

$$p_\theta\left(x^{k-1} | x^k\right) = \mathcal{N}\left(x^{k-1}; \mu_\theta\left(x^k, k\right), \Sigma_\theta\left(x^k, k\right)\right)\ , \tag{4}$$

where $\Sigma_\theta\left(x^k, k\right)$ is set to $\sigma_k^2 \mathbf{I}$ and

$$\mu_\theta\left(x^k, k\right) = \frac{1}{\sqrt{\alpha_k}}\left(x^k - \frac{1 - \alpha_k}{\sqrt{1 - \bar{\alpha}_k}}\ \epsilon_\theta\left(\sqrt{\bar{\alpha}_k}\ x^0 + \sqrt{1 - \bar{\alpha}_k}\ \epsilon, k\right)\right)\ , \tag{5}$$

where $\epsilon_\theta$ is a trainable function, parameterized by $\theta$ that predicts the noise $\epsilon$ of $x^k$ that is added during the forward process. $\epsilon_\theta$ can be trained via the simplified objective function (Ho et al., 2020)

$$\mathcal{L} = \left\| \epsilon - \epsilon_\theta\left(x^k, k\right) \right\|^2\ . \tag{6}$$

Once trained, a sample can be generated from Gaussian noise by iteratively denoising the input $K$ times (see e.g., Ho et al. (2020)).

## 2.3 CONDITIONING DIFFUSION MODELS FOR TIME SERIES DATA

Several methods exist that include conditions to generate a certain output. Class-agnostic (Nichol & Dhariwal, 2021) or text-based conditioning (Podell et al., 2024) is often used for computer vision tasks, where a label or text prompt is given, describing the object or scene to generate. The objective function in Eq. (6) at diffusion step $k$ can be supplemented with a conditioning vector $c$

$$\mathcal{L} = \left\| \epsilon - \epsilon_\theta\left(x^k, k, c\right) \right\|^2\ . \tag{7}$$

Non-categorical conditioning often makes use of alternative ideas on how to tailor a model towards a desired output.

In their work on Conditional score-based diffusion models (CSDI), Tashiro et al. (2021) combined a diffusion model with time series imputation. Their approach involves randomly masking portions of a sample, requiring the model to estimate the missing values. To guide the model towards reconstructing the masked sections, the authors utilize the unmasked parts of the signal as a condition. This condition remains uncorrupted by noise, allowing the model to leverage information from neighboring values for accurate imputation effectively. Based on CSDI, Chen et al. (2023b) introduced a novel masking strategy for imputation in IMDiffusion, specifically designed for time series anomaly detection. Their method ensures that all data points in the time series are imputed by employing two imputation instances along with two complementary masks. The resulting instances are merged to achieve a complete reconstruction of the sample, thereby enhancing the model's ability to detect anomalies more precisely.

Shen & Kwok (2023) propose TimeDiff that integrates two distinct conditioning strategies to enhance time series forecasting. During the training phase, they combine past information with the future ground truth, utilizing the resulting latent vector as the primary component of the condition. At the inference stage, the model relies solely on past time steps. Additionally, a linear autoregressive model provides a preliminary approximation of future time steps, serving as a secondary condition to further refine the model's predictions.

DiffusionAE (Pintilie et al., 2023) is an anomaly detection model that combines autoencoders and diffusion models. An autoencoder reconstructs an input signal and passes the reconstruction to the diffusion model. The authors showed that the model is more robust to small noise levels since the non-optimal autoencoder reconstruction can be seen as a slightly corrupted input, further perturbed throughout the forward process. During anomaly detection, the signal is corrupted in fewer forward steps than during training to retain information about the original signal.

The incorporation of latent vectors is a different approach for directing the generation process of diffusion models. Rasul et al. (2021) showed the beneficial impact of conditioning when doing short-term time series forecasting. They utilized an additional RNN to capture the temporal dependencies of previous time steps and integrated the updated hidden state in their model. In MG-TSD, Fan et al. (2024) refined this method by using various granularity levels of the signal as input.

Most approaches for time series forecasting condition their generative models on observed values of past time steps or their latent representation. Instead, Kollovieh et al. (2023) employ a self-guidance mechanism that allows sampling from a class-agnostic distribution during the backward process.

Besides the presented mechanisms, self-conditioning (Chen et al., 2023a) is often used in diffusion models, as it can be applied without external sources. The model is directly conditioned by its previous estimate of $\hat{x}^0$. To ensure that the model does not focus entirely on self-conditioning, it is zeroed out with 50 % probability.

All these approaches have in common that the degeneration of the sample relies on the forward process introduced in Ho et al. (2020). Our approach reinvents this step by taking the spatio-temporal correlations of time series into account.

## 3 GUIDED DEGRADATION PROCESS

While previous approaches have predominantly focused on altering the denoising process to influence the diffusion model's outcome, our methodology innovatively manipulates the forward process. This adjustment facilitates faster convergence during training and enhances robustness during inference. Our forward process corrupts the sample while preserving the temporal relationships within the signal. As demonstrated by Bansal et al. (2023), the diffusion process need not be strictly Gaussian and can be realized through various mechanisms. However, the options for guided degradation of time series data remain limited.

We propose STDM, a novel forward process for time series diffusion models that employs convolutional operators to corrupt the input signal. This forward process is defined as a Markov chain, akin to traditional diffusion models. The sample for the subsequent diffusion step $k + 1$ is obtained autoregressively by convolving the current sample $x^k$ with a fixed Gaussian kernel $H^{*1}$ and corrupting the smoothened sample with diffusion step specific noise $\boldsymbol{\epsilon}^k \sim \mathcal{N}_k(\mathbf{0}, \mathbf{I})$

$$x^{k+1} = H^{*1} * x^k + b_k \boldsymbol{\epsilon}^k \tag{8}$$

where $*$ denotes the convolution operator, and $b_k \in [0, 1]$ is predefined by a scheduler, controlling the impact of the noise on the signal. Unfolding the recursion yields

$$x^{k+1} = H^{*(k+1)} * x^0 + \sum_{j=0}^{k} H^{*(k-j)} * \left(b_j \boldsymbol{\epsilon}^j\right) \ , \tag{9}$$

where $H^{*k}$ represents the combined kernel

$$H^{*k} = \underbrace{H^{*1} * H^{*1} * ... * H^{*1}}_{k \text{ times}} \tag{10}$$

and $H^{*0} = \mathbf{I}$. We chose the initial kernel $H^{*1} = [h_{-1}, h_0, h_1]$ to be of length $l^1 = 3$. For the first diffusion steps, the influence of the kernel is local, as just neighboring time steps in the time series impact the values. With increasing $k$, the kernel size expands and captures global dependencies. The discrete convolution can be expressed as a matrix multiplication when $H^*$ is converted into the Toeplitz matrix

$$\bar{H} = \begin{bmatrix} h_0 & h_1 & & & & \\ h_{-1} & h_0 & h_1 & & & \\ & & & & & \\ & & \ddots & & & \\ & & & h_{-1} & h_0 & h_1 \\ & & & & h_{-1} & h_0 \end{bmatrix} . \tag{11}$$

To minimize the computational overhead during training and inference, we compute the kernel $H^{*k}$ for every $k$ beforehand. The expectation of $x^{k+1}$ given the initial signal $x^0$ is

$$\mathbb{E}\left(x^{k+1} \mid x^0\right) = \mathbb{E}\left(\bar{H}^{k+1} x^0 + \sum_{j=0}^{k} \bar{H}^{k-j} b_j \boldsymbol{\epsilon}^j \;\middle|\; x^0\right) \tag{12}$$

$$= \bar{H}^{k+1} x^0 , \tag{13}$$

where $\bar{H}^k$ corresponds to $H^{*k}$, which leads to the conditional distribution

$$q\left(x^k \mid x^0\right) = \mathcal{N}\left(x^k; \bar{H}^k x^0, \sum_{j=0}^{k-1} \bar{H}^{2j} b_{k-j}^2\right) . \tag{14}$$

The derivation of the covariance matrix is detailed in Appendix A.

Hence, the forward diffusion process can be written very similar to the ordinary diffusion process as

$$x^k = H^{*(k-1)} * x^0 + w^{k-1} , \tag{15}$$

where $w^k \sim \mathcal{N}\left(\mathbf{0}, \sum_{j=0}^{k-1} \bar{H}^{2j} b_{k-j}^2\right)$ is a Gaussian noise, which is spatially and temporally correlated.

As the dimensions through $x^{1:K}$ must stay the same, padding is needed. We chose a reflective padding strategy on $x^0$, as border effects are reduced. For a large $k$, the size of $H^*$ increases up to $l^{K-1} = 2(K-1) + l^1$, necessitating extensive padding. To mitigate the computational burden, we truncate the tails of the kernel, as their contributions are minimal.

The difference between STDM and DDPM is displayed in Figure 1. A time series $x^0$ is gradually corrupted at different diffusion steps $k$. In DDPM, the proportion of Gaussian noise increases with $k$ (as shown in Eq. 3), leading to a noisy signal characterized by abrupt changes in adjacent values. Conversely, our forward process (see Eq. 15) leverages correlated values during noise application. Figure 2 separately visualizes the smoothing effect of $\bar{H}$ on a single channel of an uncorrupted sample $x^0$ of length 48 from the `Solar` dataset at different diffusion steps and the resulting $x^k$. The sample is identical to that in Figure 2. The time series values are represented as a colored bar, where the color denotes the value at each time step. $x^0$ is a fine-grained time series with high variations between adjacent time steps. As $k$ evolves, the window of neighboring values influencing each time step becomes global, resulting in a more uniform bar without abrupt value changes (left side). Simultaneously, to the increasing diffusion step $k$, the influence of the original sample on the resulting $x^k$ (right side) diminishes, with the corresponding $w^k$ becoming increasingly dominant. During inference, $x^K$ is initialized as a normally distributed noise vector. Two further channels of the same sample are displayed in appendix B.

Traditional diffusion models try to predict the error $\epsilon_k$, thereby minimizing Eq. (7). However, in our approach, the error with respect to $x^0$ is according to Eq. (15) a combined, correlated noise dependent on $\bar{H}^k$. Consequently, the model must predict the entire difference between $x^k$ and $x^0$ and not only the noise level. Therefore, the training objective can be formulated as

$$\mathcal{L} = \mathbb{E}_{x^0, k, \boldsymbol{c}}\left[\left\|x^0 - \left(x^k - \epsilon_\theta\left(x^k, k, \boldsymbol{c}\right)\right)\right\|^2\right] , \tag{16}$$

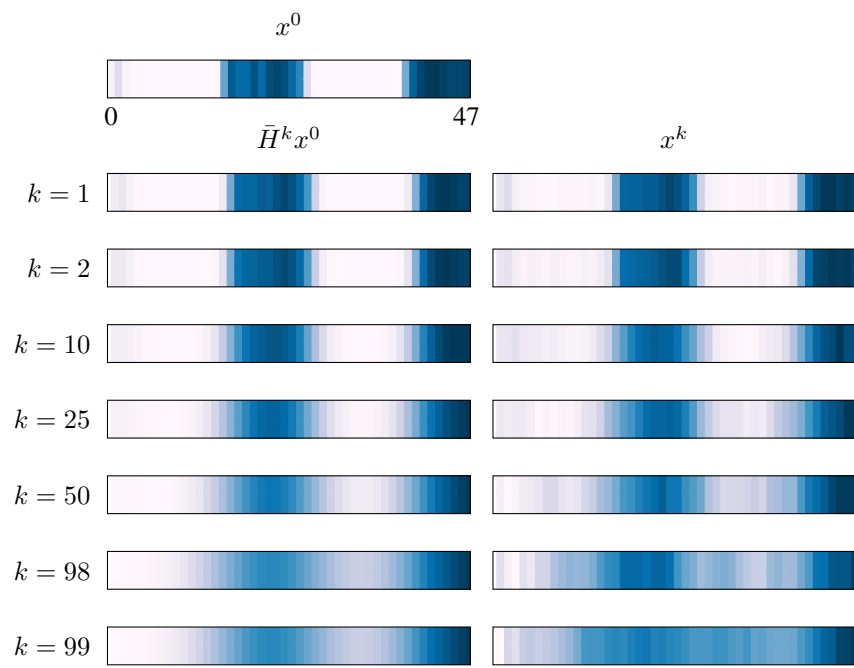

Figure 2: A single channel of a sample from the `Solar` dataset (top), the smoothing process of the sample (left), and the noisy sample at different diffusion steps $k$ (right). In this example, we used a linear scheduler for $b$.

---

**Algorithm 1** Training

1: **repeat**
2:     $x^0 \sim q(x^0)$
3:     $k \sim \text{Uniform}(\{1, ..., K\})$
4:     $w^{k-1} \sim \mathcal{N}\left(\mathbf{0}, \sum_{j=0}^{k-2} \bar{H}^{2j} b_{k-1-j}^2\right)$
5:     $x^k = H^{*(k-1)} * x^0 + w^{k-1}$
6:     $\nabla_\theta \|x^0 - \left(x^k - \epsilon_\theta\left(x^k, k, c\right)\right)\|^2$
7: **until** converged

**Algorithm 2** Sampling

1: $x^K \sim \mathcal{N}(\mathbf{0}, \mathbf{I})$
2: $w = \texttt{get\_all\_w}(K)$
3: **for** $k = K, ..., 1$ **do**
4:     $\hat{\epsilon} = \epsilon_\theta\left(x^k, k, c\right)$
5:     $\hat{x}^0 = x^k - \hat{\epsilon}$
6:     **if** $k > 1$ **then**
7:        $x^{k-1} = H^{*(k-2)} * \hat{x}^0 + w^{k-2}$
8:     **end if**
9: **end for**
10: **return** $\hat{x}^0$

---

where $c$ is a model- and task-specific conditioning vector. Conditioning remains essential, as the sampling process initiates at $x^K \sim \mathcal{N}(\mathbf{0}, \mathbf{I})$, and the diffusion network lacks prior information about the original signal.

Algorithm 1 and 2 show the training and sampling process, respectively. During training, $k$ is drawn randomly from the univariate distribution, and $w^{k-1}$ is a new Gaussian distribution in every iteration. During inference, the sample-specific $w$ for every $k$ can be computed beforehand.

## 4 EXPERIMENTS

### 4.1 BASELINES

We trained the different models on a single Nvidia RTX-4090 GPU. The implementation of the used baseline approaches can be found online at the author's Github:

- DiffusionAE: https://github.com/fbrad/DiffusionAE/
- TimeGrad: https://github.com/zalandoresearch/pytorch-ts/

Table 1: Anomaly detection results (bigger is better) of DiffusionAE with the traditional degradation process (-DDPM) and the modified version with convolution-based degradation (-STDM). The values represent the $F1_K$-AUC and $ROC_K$-AUC of 5 independent runs. The best values for each dataset are displayed in bold.

| | DiffusionAE-DDPM | | DiffusionAE-STDM | |
|---|---|---|---|---|
| | $F1_K$-AUC | $ROC_K$-AUC | $F1_K$-AUC | $ROC_K$-AUC |
| Global | $0.883_{\pm 0.003}$ | $\mathbf{0.985_{\pm 0.003}}$ | $\mathbf{0.900_{\pm 0.005}}$ | $0.982_{\pm 0.003}$ |
| Contextual | $0.777_{\pm 0.005}$ | $\mathbf{0.915_{\pm 0.003}}$ | $\mathbf{0.793_{\pm 0.022}}$ | $0.913_{\pm 0.009}$ |
| Seasonal | $0.946_{\pm 0.004}$ | $0.996_{\pm 0.001}$ | $\mathbf{0.954_{\pm 0.001}}$ | $\mathbf{0.996_{\pm 0.002}}$ |
| Shapelet | $0.685_{\pm 0.045}$ | $0.928_{\pm 0.011}$ | $\mathbf{0.749_{\pm 0.017}}$ | $\mathbf{0.948_{\pm 0.003}}$ |
| Trend | $0.530_{\pm 0.069}$ | $0.882_{\pm 0.016}$ | $\mathbf{0.698_{\pm 0.095}}$ | $\mathbf{0.923_{\pm 0.008}}$ |

To ensure a fair comprehension, we did not change hyperparameters affecting the model structure, like input sample size, layer depth, or embedding strategy. Also, preprocessing steps, like normalization and warm-up strategies, and post-processing, like adaption of metrics, remain untouched. Instead, we adjusted the number of forward and backward steps and the variance scheduler limits as the degradation procedure differed. We chose a linear scheduler with $b_1 = 0.05$ and $b_K = 0.20$.

## 4.2 RESULTS

To validate our process, we evaluated state-of-the-art diffusion-based methodologies for anomaly detection and time series forecasting. We compare the outcomes of the original implementations with our novel technique. The details of the used datasets are listed in appendix C. Anomaly detection and time series forecasting come with different metrics. We use the typical scores for each discipline. We refer to appendix D for a detailed explanation of the metrics.

**Anomaly Detection.** Pintilie et al. (2023) generated synthetic multivariate datasets to assess DiffusionAE, each embodying distinct anomaly types as delineated by NeurIPS-TS (Lai et al., 2021). Their diffusion model's efficacy was quantified using $F1_K$-AUC and $ROC_K$-AUC metrics. The $F1_K$ score computation adhered to the *PA%K* protocol (Kim et al., 2022), which employs point adjustment as utilized by Su et al. (2019) for varying *K%* of the time steps within an anomaly being detected correctly. To ensure threshold independence, ROC curves were generated for multiple thresholds *K*. The metrics in Table 1 represent the mean area under the curve (AUC) for $F1_K$ and $ROC_K$ across five independent runs for each dataset. According to Pintilie et al. (2023), the sampling process must not begin at the final diffusion step with Gaussian noise, as stated in algorithm 2, but starts at an intermediate step depending on the dataset.

In our comparative analysis of the two sampling methodologies, our convolution-based sampler consistently outperforms the traditional diffusion process in $F1_K$-AUC scores across all five synthetic datasets. The $ROC_K$-AUC scores remain predominantly high and stable. Notably, our approach demonstrates a remarkable enhancement in the dataset characterized by shape-based anomalies, achieving a relative improvement of $9.4\%$ in anomaly detection performance ($F1_K$-AUC). Furthermore, our approach significantly boosts the detection scores by $31.8\%$ for the trend-based dataset, where the traditional method exhibits notable deficiencies. However, the detection results are still lower and more volatile than those of the other datasets, which is traceable to a more accurate reconstruction of the anomalous segments Pintilie et al. (2023). Figure 3 shows a channel of two samples of the Trend dataset. Our approach (blue line) could not reconstruct the anomalous segments, which are highlighted in red. An imperfect reconstruction indicates an anomaly.

**Forecasting.** We tested our degeneration algorithm on five open-source forecasting datasets. The efficiency of a model for probabilistic time series forecasting is commonly evaluated using $CRPS_{sum}$ (Continuous Ranked Probability Score) after Salinas et al. (2019) and $NRMSE_{sum}$ (Normalized Root Mean Squared Error). Table 2 compares the standard DDPM-based method of Rasul et al. (2021) with our novel approach. Their forecasting algorithm handles every time step of a time series separately. The degradation is on a feature level. We kept this setting and applied our forward process to the feature dimension, assuming the channels are also correlated. Therefore, the results of the forecasting datasets are mixed. With STDM applied, TimeGrad could enhance the forecasting results in

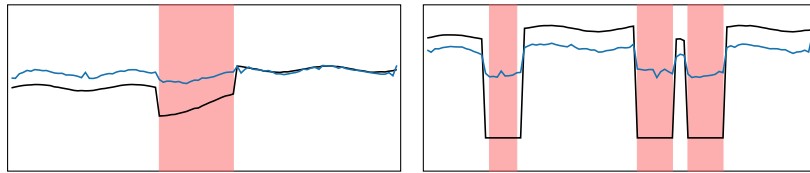

Figure 3: The sample from the Trend dataset (black) and the reconstruction (blue). The anomaly label is visualized in red.

Table 2: Forcasting results (smaller is better) of TimeGrad and MG-TSD with the traditional degradation process (-DDPM) and the modified version with convolution-based degradation (-STDM). The values represent the $\text{CRPS}_{\text{sum}}$ and $\text{NRMSE}_{\text{sum}}$ of 10 independent runs. The best values for each dataset are displayed in bold.

| | TimeGrad-DDPM | | TimeGrad-STDM | |
|---|---|---|---|---|
| | $\text{CRPS}_{\text{sum}}$ | $\text{NRMSE}_{\text{sum}}$ | $\text{CRPS}_{\text{sum}}$ | $\text{NRMSE}_{\text{sum}}$ |
| Solar | $0.3744_{\pm 0.0448}$ | $0.7454_{\pm 0.0776}$ | $\mathbf{0.2956}_{\pm\mathbf{0.0486}}$ | $\mathbf{0.6222}_{\pm\mathbf{0.1090}}$ |
| Electricity | $\mathbf{0.0222}_{\pm\mathbf{0.0013}}$ | $\mathbf{0.0402}_{\pm\mathbf{0.0025}}$ | $0.0453_{\pm 0.0110}$ | $0.0620_{\pm 0.0227}$ |
| Traffic | $0.0431_{\pm 0.0081}$ | $0.0795_{\pm 0.0300}$ | $\mathbf{0.0420}_{\pm\mathbf{0.0101}}$ | $\mathbf{0.0692}_{\pm\mathbf{0.0234}}$ |
| Taxi | $0.1265_{\pm 0.0100}$ | $0.2339_{\pm 0.0176}$ | $\mathbf{0.1214}_{\pm\mathbf{0.0224}}$ | $\mathbf{0.2270}_{\pm\mathbf{0.0194}}$ |

most of the investigated datasets. The diffusion model encounters the most challenges when applied to the Solar dataset with both forward processes.

## 5 CONCLUSION

In this paper, we introduced STDM, a novel technique to guide the forward process in diffusion models specifically tailored for time series data. Our approach involves the smoothing of the signal, which is subsequently corrupted by Gaussian noise. The noises at various stages exhibit correlations and spatial dependencies across their dimensions. The forward process to any arbitrary intermediate diffusion step can be computed in a single step. Given that our novel forward process mirrors the structure of the conventional diffusion process, it can be seamlessly integrated with existing diffusion models.

Our experiments underscore the efficacy of our approach, demonstrating enhancements in anomaly detection and forecasting tasks with minimal effort. For a fair comparison, a model-specific adjustment of parameters is out of scope but we believe that further hyperparameter optimizations hold the potential to yield even more impressive results. This work paves the way for future research and applications, offering a robust framework for improving the performance of diffusion models in various time series analysis tasks.

### REPRODUCIBILITY STATEMENT

To ensure the reproducibility and completeness of this paper, we have included an appendix with additional information. In appendix C, the used open-source datasets are presented. Appendix D provides an overview of the used metrics for performance measurement. While the baseline methods are publicly available (see section 4.1), the implementation details of our algorithm can be seen in the pseudocode in section 3. Our code will be made publicly accessible once the paper is accepted.

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

## A    DERIVATION OF THE COVARIANCE MATRIX

The covariance matrix of the conditional distribution in Eq. (14) can be written as

$$\text{Cov}\left(x^{k+1}\right) = \mathbb{E}\left(x^{k+1}x^{k+1\prime}\right) - \mathbb{E}\left(x^{k+1}\right)\mathbb{E}\left(x^{k+1}\right)' \tag{17}$$

$$= \mathbb{E}\left[\left(\bar{H}x^k + b_k\epsilon_k\right)\left(\bar{H}x^k + b_k\epsilon_k\right)'\right] - \mathbb{E}\left(\bar{H}x^k + b_k\epsilon_k\right)\mathbb{E}\left(\bar{H}x^k + b_k\epsilon_k\right)' \tag{18}$$

$$= \bar{H}\,\mathbb{E}\left(x^k x^{k\prime}\right)\bar{H}' + \mathbb{E}\left(\bar{H}x^k b_k\epsilon_k'\right) + \mathbb{E}\left(b_k\epsilon_k x^{k\prime}\bar{H}'\right) + \mathbb{E}\left(b_k^2\epsilon_k\epsilon_k'\right) - \bar{H}\,\mathbb{E}\left(x^k\right)\mathbb{E}\left(x^{k\prime}\right)\bar{H}' \tag{19}$$

$$= \bar{H}\,\text{Cov}\left(x^k\right)\bar{H}' + b_k^2\mathbf{I} \tag{20}$$

## B    MULTIVARIATE FORWARD PROCESS

Figure 4 visualizes a multivariate version of the degradation process. There are two channels displayed for every forward diffusion step.

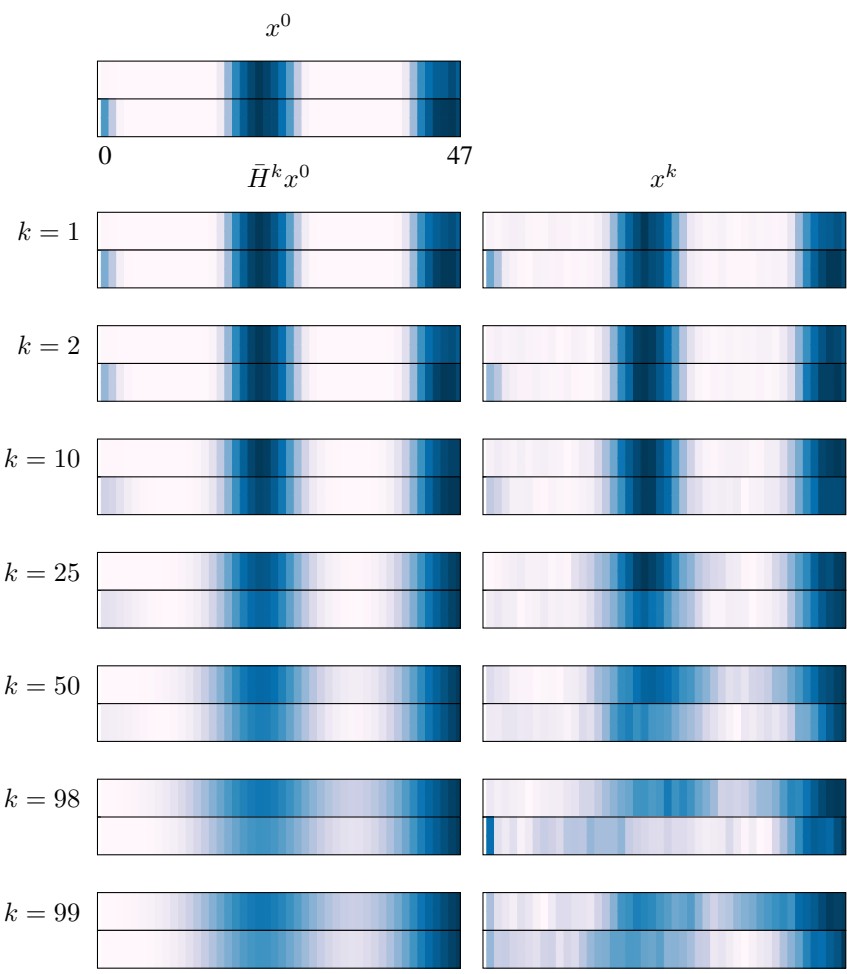

Figure 4: Two channels of a sample from the `Solar` dataset (top), the smoothing process of the sample (left), and the noisy sample at different diffusion steps $k$ (right). In this example, we used a linear scheduler for $b$.

# C  DATASETS

## C.1  ANOMALY DETECTION

The synthetic datasets listed in Table 1 can be obtained via the project website of the DiffusionAE algorithm: `https://github.com/fbrad/DiffusionAE/`. The main dataset statistics are listed in Table 3. Each synthetic dataset has five dimensions and anomalies in train and test data. For more details, we refer to Pintilie et al. (2023).

Table 3: Information about the synthetic datasets for anomaly detection task

| Name | Anomaly Type | Dimensions | Train | Val | Test |
|------|------|------|------|------|------|
| Global | point | 5 | 20000 | 10000 | 20000 |
| Contextual | point | 5 | 20000 | 10000 | 20000 |
| Seasonal | pattern | 5 | 20000 | 10000 | 20000 |
| Shapelet | pattern | 5 | 20000 | 10000 | 20000 |
| Trend | pattern | 5 | 20000 | 10000 | 20000 |

## C.2  FORECASTING

For the time series forecasting benchmark datasets, we use `Solar`, `Electricity`, `Traffic`, and `Taxi`.They can all be obtained via the *GluonTS* library (Alexandrov et al., 2020). The details are listed in Table 4.

Table 4: Information about the datasets for forecasting task

| Name | Number of Series | Frequency | Context Length | Prediciton Length |
|------|------|------|------|------|
| Solar | 137 | hour | 24 | 24 |
| Electricity | 370 | hour | 24 | 24 |
| Traffic | 963 | hour | 24 | 24 |
| Taxi | 1214 | 30 min | 24 | 24 |

# D  METRICS

## D.1  ANOMALY DETECTION

The F1 score is a reliable indicator of the accuracy of anomaly detection algorithms. It can be calculated by determining the number of true positives (TPs), false negatives (FNs), and false positives (FPs). Regarding the evaluation strategy, the numbers can vary. For detection on a time-step level, every time-step is categorized depending on the anomaly score as normal or anomalous. With the point-adjustment strategy (Su et al., 2019), the categorization of every time step can be adapted in a post-processing step: Every time step within an anomalous segment is considered as abnormal as long as one time step is categorized correctly, which leads to higher F1 scores. Generally, the F1 score can be formulated as the harmonic mean between precision and recall

$$\text{F1} = 2\,\frac{\text{P R}}{\text{P} + \text{R}}\ , \tag{21}$$

where precision P is

$$\text{P} = \frac{\text{TP}}{\text{TP} + \text{FP}}\ , \tag{22}$$

and recall R is

$$\text{R} = \frac{\text{TP}}{\text{TP} + \text{FN}}\ . \tag{23}$$

For fairer comparison, Pintilie et al. (2023) calculated the F1 scores for different percentages of abnormal time steps in an anomaly segment being detected correctly. The $\text{F1}_K$-AUC results in Table 1 are the area under the curve for the different F1 scores.

The threshold categorizing time steps as normal or abnormal is crucial for a high overall detection result, the true positive rate (TPR) and false positive rate (FPR), which can be written as

$$\text{TPR} = \frac{\text{TP}}{\text{TP} + \text{FN}} \tag{24}$$

and

$$\text{FPR} = \frac{\text{FP}}{\text{FP} + \text{TN}} \ . \tag{25}$$

To eliminate this dependency, the ROC was determined for different threshold values as

$$\text{ROC} = 0.5 \ (1 + \text{TPR} - \text{FPR}) \ . \tag{26}$$

The area under the different ROC values is represented in Table 1 as $\text{ROC}_K$-AUC

### D.2 FORECASTING

The metrics for forecasting are pre-implemented in the *GluonTS* library (Alexandrov et al., 2020). We summarize the used ones as follows.

**CRPS$_{\text{sum}}$:** CRPS is a frequently used metric for probabilistic forecasting methods, introduced by Matheson & Winkler (1976). It measures the compatibility of a cumulative distribution function $F$ with the observation $y$ as

$$\text{CRPS}\,(F, y) = \int_{\mathbb{R}} \left(F(x) - \mathbb{1}\,(x \geq y)\right)^2 \mathrm{d}x \ , \tag{27}$$

where $\mathbb{1}$ is the Heaviside step function. Salinas et al. (2019) extended the score to CRPS$_{\text{sum}}$ for multivariate time series data as

$$\text{CRPS}_{\text{sum}} = \mathbb{E}_t \left[ \text{CRPS} \left( F_{\text{sum}}^{-1}, \sum_i x_t^i \right) \right] \ , \tag{28}$$

where $F_{\text{sum}}^{-1}$ is obtained by first summing samples across dimensions and then sorting to get quantiles.

**NRMSE:** NRMSE represents the normalized version of the Root Mean Squared Error. According to Fan et al. (2024) it can be written as

$$\text{NRMSE} = \sqrt{\frac{\text{mean}\left(\left(\hat{Y} - Y\right)^2\right)}{\text{mean}\left(|Y|\right)}} \ , \tag{29}$$

where $\hat{Y}$ is the predicted time series and $Y$ is the target.

