# OpenReview forum: "STDM: Spatio-Temporal Diffusion Models for Time Series Analysis"
_ICLR.cc/2025/Conference — Submitted to ICLR 2025_

### Official Review · Reviewer_G7bm · 2024-10-23

**Soundness:** 1
**Presentation:** 2
**Contribution:** 2
**Rating:** 3
**Confidence:** 4

**Summary:**

The author proposed Spatio-Temporal Diffusion Models (STDM), introducing a new forward process for time series diffusion models. The new forward process tries to use step-dependent convolutional kernels for capturing spatial relations and a combined, correlated noise to degenerate the input. The method can be integrated seamlessly into existing time series models like DiffuisonAE and TimeGrad, replacing the original forward process. Experiment results show the effectiveness of the proposed method on two tasks: time series anomaly detection and forecasting, with one baseline model examined for each task.

**Strengths:**

The idea of incorporating explicit capture of temporal patterns within the time series during forward process is inspiring. The step-dependent convolutional operator for executing the forward process is novel and reasonable.

**Weaknesses:**

- The eq. (16) needs to be validated, since the forward process is modified, the differences in the derivation should be noted.
- The method section seems incomplete, for example, the definition of $c$ is not clearly stated.
- The experiments are only on one baseline method for each task, which seems not adequate. The content of Table 2 is not as described in the caption (MG-TSD is mentioned in caption but not shown in table content).
- In TimeGrad, the multi-variate time series are generated autoregressively, which seems to contradict with the proposed method where the  $x^0$ denotes a multi-step series. It's not clear to me how the convolution kernel is applied on cross-sectional data (containing only one time step). Please correct me if I misunderstood some steps here.

**Questions:**

- Could author provide a step-by-step derivation of equation (16), highlighting how it differs from the traditional diffusion process derivation?
- Could author provide a clear definition of $c$, regarding each of the evaluated tasks?
- How is the proposed method applied on both autoregressive and non-autoregressive generation process? Particularly, how it works with TimeGrad?

---

### Official Review · Reviewer_QdZU · 2024-10-27

**Soundness:** 2
**Presentation:** 2
**Contribution:** 2
**Rating:** 3
**Confidence:** 4

**Summary:**

This work introduce a novel approach to enhance denoising diffusion models for time series tasks, addressing the challenge of conditioning for accurate reconstruction and sampling based on past time steps. Unlike existing methods, STDM guides the diffusion model's forward process by leveraging the high correlation between neighboring time steps. This is achieved through a diffusion step-dependent convolutional kernel and correlated noise to capture spatial relations and refine the degradation of inputs.

**Strengths:**

Overall, the writing is fluent and easy to follow. The key details are well-explained. The paper replaces the traditional linear transformation in the noise addition process of diffusion models with convolution operations, which, to my knowledge, has not been seen in other work.

**Weaknesses:**

The motivation of the paper is not very clear. From the perspective of guided diffusion, the conditioning approach used here doesn’t seem different from existing works. In my view, the main contribution of this paper lies in the use of convolution operations in the noise addition process, which introduces a smoothing effect on the signal distinct from traditional diffusion models. However, this smoothing approach doesn’t appear particularly meaningful, as in diffusion models we generally don’t focus much on the intermediate states in the noise/denoising process but rather only on the final generated samples. Additionally, the experiments are weak, as the paper only compares against the original DDPM and overlooks recent work from the past few years.

**Questions:**

1. Motivation issue. See weaknesses. Does using convolution-based noise addition/removal versus Gaussian-based noise addition/removal have a substantial impact on sample generation? Can this be theoretically proven?

2. Eq(16). If I understand correctly, $x_0$ should be $x_{k-1}$.

3. Eq(14). As $k \to \infty$, will this distribution converge to $N(0, I)$? This is relevant because in the experiments, you directly sample $x_K \sim N(0, I)$.

4. The experiments are too simplistic. I recommend adding more baselines to compare with diffusion models that use different noise addition processes.

---

### Official Review · Reviewer_xmm8 · 2024-10-27

**Soundness:** 1
**Presentation:** 1
**Contribution:** 1
**Rating:** 3
**Confidence:** 5

**Summary:**

This paper proposes a Spatio-Temporal Diffusion Models for generating entire samples of time series. Experiments are carried on synthetic and real-world datasets.

**Strengths:**

The motivatin of producing entirely new sample via diffussion model seem to be interesting.

**Weaknesses:**

1. Figure 1 is clear to illustrate the strength of STDM in comparison with vanilla DDPM.
2. What does ``Spatio-Temporal'  mean and is related to the proposed approach?
3. More relevant works are needed to be discussed and compared, including Csdi: Conditional score-based diffusion models for probabilistic time series imputation (NIPS 2021); Self-Supervised Learning of Time Series Representation via Diffusion Process and Imputation-Interpolation-Forecasting Mask (KDD 2024)
4. The contribution and novelty are unclear. What is the superiority of STDM in comparison with current time series SSL methods?
5. Vital baselines are missed, e.g., SimMTM (NIPS 2023), TS-TCC (TPAMI 2024), TS2Vec (AAAI2022)......
6. More datasets should be analyzed, e.g., ETTh1/h2/m1/m2 for time series forecasting, and SMD/SWAT for anomaly detection

**Questions:**

Please refer to weankess

---

### Official Review · Reviewer_YX33 · 2024-11-03

**Soundness:** 2
**Presentation:** 1
**Contribution:** 2
**Rating:** 3
**Confidence:** 4

**Summary:**

This paper proposes Spatio-Temporal Diffusion Model (STDM) which redesigns the diffusion forward process for capturing correlations in time series data, and can be seamlessly integrated into current diffusion models to improve their performance in time series analysis tasks. Experiments explore the performence of STDM in time series anomaly detection and forecasting tasks.

**Strengths:**

- Novel research perspective. As far as the reviewer is aware, this is the first paper to improve the performance on time series analysis tasks by redesigning the diffusion forward process.
- The proposed method is flexible and extensible, and can be seamlessly integrated into the time series diffusion model to improve performance.

**Weaknesses:**

- Motivation is not clear and does not tell a coherent story. I do not understand the motivation and significance of paying attention to the temporal patterns in the diffusion forward process of adding noise. It appears that the temporal correlations introduced by the noise in the forward process may enable the model to effectively consider and learn these correlations for denoising in the reverse process. However, the writing of the paper does not clearly explain this.
- In Section 3, the author mentioned that "our methodology innovatively manipulates the forward process. This adjustment facilitates faster convergence during training and enhances robustness during inference". Nevertheless, the mechanism by which STDM accelerates training and improves inference robustness are not sufficiently explained, and both theoretical analysis and lacks empirical evidence to support this assertion.
- The experiment results only evaluate the DiffusionAE and TimeGrad models, which are not enough to support the effectivenenss of the proposed method. And there is a notable absence of baselines for time series forecasting and anomaly detection, which limits the comprehensiveness of the evaluation.
- The writing and charts are extremely crude and rudimentary.

**Questions:**

- Is convolution kernel $H$ trainable? What is the extra training cost of this design in diffusion forward process?
- How does the convolution kernel capture spatio-temporal correlations? In my opinion, kernel $H$ seems to be able to capture only **temporal** pattern correlations within a series, but the author claims that STDM captures **spatial** correlations (for example, the contribution section). The method cannot capture spatio-temporal correlations of multivariate series anyway, I don't understand why the author named it **Spatio-Temporal** diffusion model.

---

### Meta-Review · Area_Chair_i6zF · 2024-12-27

**Metareview:**

The paper proposes a new forward process for diffusion models of time series that leverages convolution. The method can be incorporated into existing diffusion models. The method was evaluated on anomaly detection and forecasting tasks.

Strengths:
+ The paper's aim and proposed approach are interesting and novel as mentioned by the reviewers.

Weaknesses:
+ Reviewers found the motivation of the approach weak. It wasn't clear why using convolution in this setting is a good thing.
+ The method is applied only to two diffusion models, and one baseline was used for each task, which is not enough.
+ Several key baselines are missing, the paper focuses on DDPM which is somewhat old in the context of diffusion modeling, a fast-moving research area.

**Additional Comments On Reviewer Discussion:**

The authors did not provide any rebuttal. The reviewers made valid points that were not addressed.

---

### Decision · Program_Chairs · 2025-01-22

Reject